# Cryo-electron tomography structure of Arp2/3 complex in cells reveals new insights into the branch junction

Florian Fäßler [1], Georgi Dimchev[1], Victor-Valentin Hodirnau[1], William Wan[2] & Florian K. M. Schur [1✉]

The actin-related protein (Arp)2/3 complex nucleates branched actin filament networks pivotal for cell migration, endocytosis and pathogen infection. Its activation is tightly regulated and involves complex structural rearrangements and actin filament binding, which are yet to be understood. Here, we report a 9.0 Å resolution structure of the actin filament Arp2/3 complex branch junction in cells using cryo-electron tomography and subtomogram averaging. This allows us to generate an accurate model of the active Arp2/3 complex in the branch junction and its interaction with actin filaments. Notably, our model reveals a previously undescribed set of interactions of the Arp2/3 complex with the mother filament, significantly different to the previous branch junction model. Our structure also indicates a central role for the ArpC3 subunit in stabilizing the active conformation.

¹Institute of Science and Technology (IST) Austria, Klosterneuburg, Austria. ² Department of Biochemistry and Center for Structural Biology, Vanderbilt University, Nashville, United States of America. ✉email: florian.schur@ist.ac.at

Spontaneous actin filament polymerization is a kinetically unfavorable process and nucleation factors are required to overcome this rate-limiting step in actin filament assembly. The heptameric 224 kDa Arp2/3 complex, consisting of the two actin-related proteins 2 and 3 (Arp2 and Arp3), and five additional subunits ArpC1-5, generates branched actin networks by inducing the formation of so-called actin filament Arp2/3 complex branch junctions on preexisting actin filaments[1].

The Arp2/3 complex plays a role in both physiological and pathological processes, such as lamellipodial protrusion, vesicle trafficking, and also the motility of various pathogens[2]. Arp2/3 complex activation is tightly regulated, requiring multiple regulatory proteins, including nucleation promoting factors (NPFs), ATP, and monomeric actin[3]. NPFs, such as members of the Wiskott–Aldrich syndrome protein (WASP) and SCAR/WAVE families, induce the structural changes required for Arp2/3 complex activation via their WASP homology 2 (W), connector (C), and acidic (A, or combined: WCA) domains, where the CA motifs bind the Arp2/3 complex and the W motif delivers the first actin monomer[4–6].

The available structural and biochemical information of the Arp2/3 complex and its interactions with other proteins have been mostly obtained via a reductionist approach, i.e., studying the Arp2/3 complex in vitro in isolation, bound to NPFs or their fragments[7,8], or stabilizing and destabilizing proteins, including cortactin[7], coronin, or Glia maturation factor (GMF)[9,10]. Details on the conformation of the Arp2/3 complex within the branch, and its interaction with the mother and daughter filament were largely derived from fitting high-resolution x-ray crystallography models of the inactive complex[11,12] into low-resolution in vitro electron microscopy (EM) and electron tomography (ET) reconstructions[13–15], as well as molecular dynamics (MD) simulations[16]. These studies led to the model that upon activation, the Arp2 and Arp3 subunits reposition into a short-pitch conformation resembling an actin filament-like state, providing the basis for further actin polymerization. Binding of the Arp2/3 complex to the mother filament was also proposed to lead to a conformational change in the filament, increasing branch junction stability[13,16]. However, due to the low resolutions (~25 Å) of the previously determined branch junction model, the currently suggested molecular contacts of the active Arp2/3 complex in the branch junction and the associated changes in the mother filament remain ambiguous. Specifically, an accurate model of the actin filament Arp2/3 complex branch junction has been lacking; such a model would better describe the functional consequences of the exact interfaces in the branch junction on its formation and stability.

Cryo-electron tomography (cryo-ET) combined with image processing approaches, such as subtomogram averaging (STA) can provide structural insights into native cellular environments[17]. Medium to high-resolution structures of proteins within their native environment using these methods have so far been only described for a few specimens, often large or highly symmetrical assemblies[18–22]. Among the limitations that restrict the resolution that can be obtained in cells are the thickness of the specimen, reduction of which often requires additional specimen preparation steps, and the number of protein complexes available for generating a higher-resolution structure[17].

The lamellipodium is a thin sheet-like membrane extension at the leading edge of migrating cells, which has a height of 100–200 nm and is densely filled with an Arp2/3 complex-dependent branched actin filament network (also referred to as dendritic network). It has been a prominent model to study actin network topology and ultrastructure using both room-temperature (RT) and cryo-EM approaches[23–26]. Due to the large number of actin filament Arp2/3 complex branch junctions within lamellipodia, we considered them the optimal target to determine the structure of branch junctions in their native environment at the highest possible resolution, and to provide the outstanding answers to the important aforementioned open points on branch junction formation and stability.

Here we report, using cryo-ET and STA on lamellipodia of fibroblast cells, a 9 Å structure of the actin filament Arp2/3 complex branch junction. This allows us to obtain an accurate model of the active Arp2/3 complex in the branch junction, which reveals a new set of interactions within the complex and with the mother filament, different to previous models.

## Results and discussion

**Cryo-ET of actin filament Arp2/3 complex branch junctions in fibroblast lamellipodia.** We performed cryo-ET of lamellipodia of NIH-3T3 fibroblast cells, which we transiently transfected with a constitutively active variant of the small GTPase Rac (L61Rac). Rac overexpression has been previously reported to enhance lamellipodia formation by activating NPFs[24,27], facilitating the selection of appropriate regions for cryo-ET data acquisition. In order to further optimize conditions for image processing, we used established extraction and fixation protocols involving the actin-stabilizing toxin phalloidin to remove the plasma membrane, while stabilizing actin filaments, allowing for a better visualization of actin filaments and bound complexes[24,26]. In agreement with previous observations, branch junctions were clearly retained within lamellipodia after extraction and fixation (Supplementary Fig. 1 and Supplementary Movie 1).

Using a combination of semiautomatic particle detection, classification, STA, and multiparticle refinement[22], we obtained a 9.0 Å resolution structure of the branch junction from aligning and averaging 14,296 subtomograms (Fig. 1a, Supplementary Fig. 2 and Supplementary Movie 2, see Supplementary Table 1 and "Methods" for details). At this resolution, all subunits of the Arp2/3 complex and also individual actin subunits of the mother (designated here as M1–M8) and daughter filaments (D1–D3) are clearly resolved.

Moreover, visible secondary structure details corresponding to alpha-helices and various loops allowed an unambiguous placement of available models of the individual Arp2/3 complex subunits and actin filament subunits into our density, without requiring any large-scale modifications of their tertiary structure. For this, the subunits of the inactive Arp2/3 complex[12] (pdb 1TYQ) and 11 actin molecules derived from a nucleotide and phalloidin-bound, aged actin filament[28] (pdb 6T20) were all individually rigid-body fitted into the EM density. As subdomains 1 and 2 of Arp2 were not present in pdb 1TYQ, the corresponding subdomains from a GMF-bound Arp2/3 complex[9] (pdb 4JD2) were used to produce a chimeric Arp2 model. Our structure also allowed us to model the backbone trace for some regions in the Arp2/3 complex that were not resolved in crystal structures (Supplementary Table 2). We then used MD-based fitting[29] to remove steric clashes and to generate a complete actin filament Arp2/3 complex branch junction model (Fig. 1b, Supplementary Fig. 3, Supplementary Movie 3, and Supplementary Tables 2 and 3). The density corresponding to ArpC5 suggested increased flexibility at its N-terminus (Supplementary Fig. 3). This flexibility could be due to alternate conformations of this region of the subunit, caused by the existence of the two isoforms ArpC5 and ArpC5L, in line with a recent study[30]. Hence, we restrained the movements of this subunit in the MD modeling. The quality of the reconstruction was further underlined by the presence of densities not occupied by either the Arp2/3 complex or actin, but at the exact location at which phalloidin would be expected (Supplementary Fig. 3). Overall, our model reveals the conformation of in total 18 protein subunits

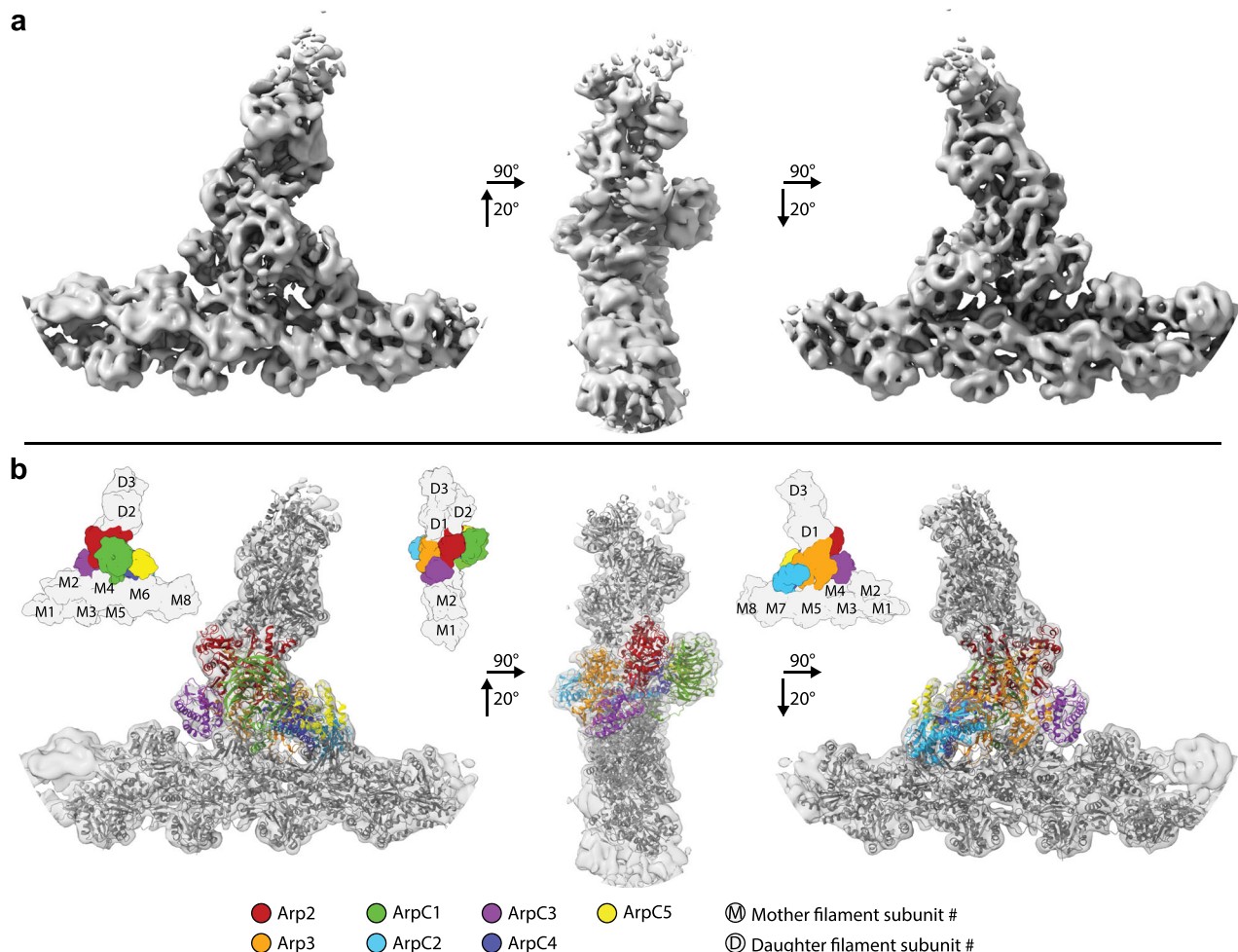

**Fig. 1 Subnanometer structure of the actin filament Arp2/3 complex branch junction in cells. a** Isosurface representation of the actin filament Arp2/3 complex branch junction in cells at 9 Å resolution. The structure is shown from three orientations. A guide for orientation is given in **b**. **b** The electron microscopy density map (shown transparent) with the flexibly fitted models of the Arp2/3 complex subunits and actin filaments. Schematic guides indicate the positions of the individual subunits of the complex, and the actin subunits of both the mother (M#) and daughter filaments (D#). Subunit colors are annotated in the figure with Arp2 being red, Arp3 orange, ArpC1 green, ArpC2 light blue, ArpC3 violet, ArpC4 dark blue, and ArpC5 yellow. Actin is shown in gray. The color code and legend are used throughout the manuscript to aid the reader.

(the heptameric Arp2/3 complex and 11 actin filament subunits); however, the resolution of our structure did not allow deriving unambiguous side chain information.

The fact that we were able to obtain subnanometer resolution in large parts of our structure, including the first actin subunits of the daughter filament (see the local resolution measurement in Supplementary Fig. 2) indicates rigidity of the branch junction. We determined the angle of the branch junction in our structure to be 71°, which is in agreement with some, but different to other previously reported in vitro and in situ studies, which showed branch junction angles ranging from 67° to ~80° (refs. [13,15,16,24,31–33]), a variation that could be partially due to potentially different approaches for angle measurement. This led us to examine in more detail the structural changes occurring in the Arp2/3 complex upon branch junction formation and to compare our model to previously suggested conformations of the branch junction[13,16,34]. Specifically, we sought to understand the contribution of the different subunits to branch junction conformation and stability, in particular their contact with both the actin mother and daughter filament.

**The structure of the active Arp2/3 complex.** First, we compared our active Arp2/3 complex model in the branch junction to a

model of the soluble inactive complex obtained by x-ray crystallography[12] (Fig. 2). The biggest change upon branch junction formation was the actin-like heterodimer arrangement of Arp2 and Arp3, which resembles the first two subunits of the daughter filament (Supplementary Fig. 4). In this regard, our structure of the activated Arp2/3 complex in branch junctions agrees with a recently published in vitro structure of the NPF Dip1-activated *Schizosaccharomyces pombe* Arp2/3 complex that generates unbranched actin filaments[35] (Supplementary Fig. 4 and Supplementary Fig. 5a, b). In both structures of the activated Arp2/3 complex, as well as in a MD simulation-derived model[36], ArpC2 and ArpC4 define the center of rotation and translation of two subcomplexes consisting of ArpC2, Arp3, and ArpC3 and ArpC4, ArpC5, ArpC1, and Arp2, respectively, that move against each other to form the short-pitch conformation (Supplementary Movies 4 and 5). In the case of unbranched filaments, nucleated by the Dip1-activated Arp2/3 complex, this structural change is induced by binding of Dip1 to ArpC4. In the branch junction, this conformation is most likely induced by the actin mother filament, which forms an extensive interface with the ArpC2 and ArpC4 subunits[13,37] (see also further description below). Indeed, the mother filament has been shown to be required for the activation of most types of the Arp2/3 complex[13,34,38], and its role

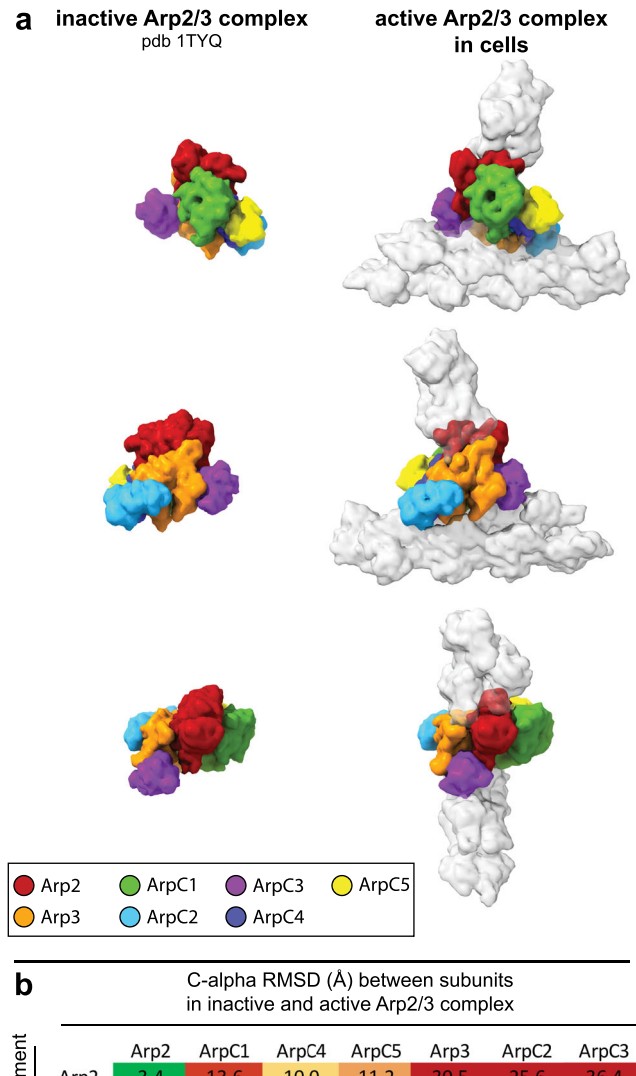

**a**  inactive Arp2/3 complex
pdb 1TYQ

active Arp2/3 complex
in cells

Arp2 ● (red)  ArpC1 ● (green)  ArpC3 ● (violet)  ArpC5 ● (yellow)
Arp3 ● (orange)  ArpC2 ● (light blue)  ArpC4 ● (dark blue)

**b**

C-alpha RMSD (Å) between subunits
in inactive and active Arp2/3 complex

subunit used for alignment

| | Arp2 | ArpC1 | ArpC4 | ArpC5 | Arp3 | ArpC2 | ArpC3 |
|---|---|---|---|---|---|---|---|
| Arp2 | 3.4 | 13.6 | 10.0 | 11.2 | 30.5 | 25.6 | 36.4 |
| ArpC1 | 8.2 | 3.7 | 4.2 | 3.0 | 12.7 | 11.0 | 17.0 |
| ArpC4 | 8.4 | 4.3 | 2.5 | 2.5 | 12.3 | 7.5 | 19.6 |
| ArpC5 | 9.3 | 6.1 | 4.2 | 0.5 | 17.8 | 13.9 | 20.1 |
| Arp3 | 22.7 | 12.8 | 8.6 | 8.7 | 5.3 | 5.5 | 9.5 |
| ArpC2 | 18.6 | 9.4 | 7.3 | 12.5 | 6.4 | 2.6 | 15.8 |
| ArpC3 | 27.4 | 24.6 | 18.2 | 24.1 | 7.1 | 12.9 | 2.5 |

**Fig. 2 Comparison of Arp2/3 complex in its inactive conformation and in the branch junction in cells. a** Molecular models of the Arp2/3 complex in the inactive (derived from pdb 1TYQ) and the active conformation shown as density maps filtered to 9.5 Å resolution. The models are shown from three orientations, corresponding to the views in Fig. 1. Subunit colors are annotated in the figure with Arp2 being red, Arp3 orange, ArpC1 green, ArpC2 light blue, ArpC3 violet, ArpC4 dark blue, and ArpC5 yellow. Actin is shown in gray. **b** RMSD values (in Å) calculated between the inactive and active conformations of the individual Arp2/3 subunits (based on the models used in **a**). Rows indicate which subunit was used for aligning the full models against each other, prior to measurements between individual subunits of the inactive and active Arp2/3 complexes (indicated in the columns). The RMSD analysis reveals that the structural transition upon complex activation is accommodated by two subcomplexes consisting of Arp2, ArpC1, ArpC4 and ArpC5 and Arp3, ArpC2 and ArpC3, respectively, that rotate against each other along an axis formed by the large helices of ArpC2 and ArpC4 (Supplementary Movie 4). RMSD variations between the same subunits in the inactive and active conformations derive from changes upon MD-based modeling of the x-ray crystal structure-derived model into the electron microscopy density map of the branch junction.

in promoting the conformational changes in the Arp2/3 complex to reposition Arp2 has also been suggested in a study, using steered MD simulations[36].

The structure of the activated Arp2/3 complex in the branch junction also reveals differences in inter-subunit contacts and distances compared to the Dip1-activated complex (Supplementary Fig. 5a, b). Specifically, in the branch junction, we observe a significant movement of ArpC3 toward the D-loop of Arp2 providing additional stabilization to the active conformation (Fig. 2a, b and Supplementary Fig. 5b).

**Actin filament Arp2/3 complex interactions.** The currently available structural information on Arp2/3 complex interactions with the mother filament within the branch junction is predominantly based on the model derived from a negative stain ET structure of in vitro assembled branch junctions[13], which subsequently was refined using all atom simulations[16] (Supplementary Fig. 5a (right), c). The 2.6 nm resolution of the in vitro EM reconstruction correctly suggested the position of Arp2 next to Arp3 in a short-pitch conformation. However, it did not reveal the other conformational changes associated with the rotation and translation of the subcomplexes in Arp2/3 complex (Supplementary Movies 4 and 5), described by others[35,36] and us. Given the improved resolution of our structure, we sought to compare our model to the previously published model[13,16] to provide a clearer description of Arp2/3 complex mother filament binding.

In the previous model of the branch junction, fitting the mother filament into the 2.6 nm resolution reconstruction suggested that changes in the conformation of two mother filament subunits (termed M4 and M6 in our model) are required to firmly anchor the complex to the mother filament[13].

While, rigid-body fitting a model of filamentous actin (pdb 6T20)[28] into the mother filament density of our 9 Å branch junction structure already showed good correspondence, it was improved by refining the fit of the individual actin molecules. This led to small changes from their original filamentous conformation (Supplementary Fig. 6a, b), but did not affect the conformation of individual actin filament subunits (Supplementary Fig. 6c), in contrast to what was suggested by the low-resolution EM structure of the branch junction[13]. Specifically, mother actin filament subunits M4 and M6, which were supposed to adapt their conformation, do not change (1.8 and 2.2 Å RMSD between C-alpha atoms of M4 and M6 subunits in the mother filament, respectively and pdb 6T20). M4 forms only very few contacts with the Arp2/3 complex within our model of the branch junction.

The previous model[13] suggested that all seven Arp2/3 complex subunits contact the mother filament. To derive a more accurate description of interaction interfaces between the Arp2/3 complex and the mother actin filament in our model, we probed for C-alpha atoms of Arp2/3 complex subunits and actin filament subunits being in vicinity closer than 10 Å (Fig. 3 and Supplementary Table 4). Importantly, in our model not all Arp2/3 complex subunits, but only five subunits, namely Arp3, ArpC1, ArpC2, ArpC3, and ArpC4, contact the mother filament (Fig. 3a and Supplementary Table 4).

Correspondingly, in total five actin subunits (M2, M4–M7) in the mother actin filament are positioned to contact the Arp2/3 complex. A prominent interaction surface between the complex and the mother filament is formed by ArpC2 and ArpC4 contacting M5, M7, and M6, respectively (Fig. 3a, b). This is in agreement with biochemical evidence showing the importance of some of the residues involved in this interaction to nucleation and branch junction stability[34,37]. The ß-propeller domain of ArpC1

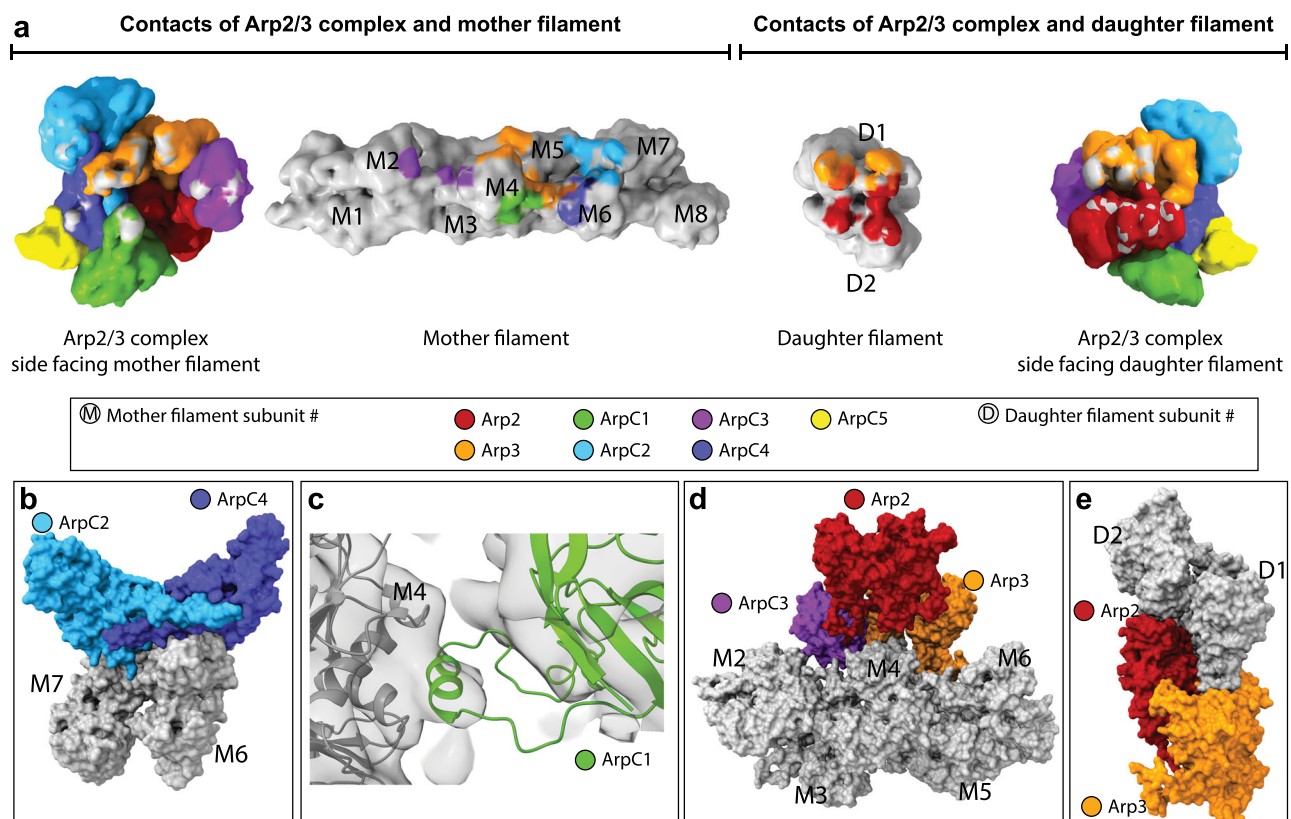

**Fig. 3 Actin–Arp2/3 complex interaction surfaces within the branch junction. a** Interaction surfaces between the Arp2/3 complex, and the actin mother and daughter filament. The surfaces for the Arp2/3 complex, mother and daughter filament are shown as density maps at 9.5 Å resolution generated from their respective models. Gray coloring on the surface of Arp2/3 subunits indicates contact sites with actin and coloring of actin subunits in a specific color indicates a contact site with the associated Arp2/3 subunit. Subunit colors are annotated in the figure with Arp2 being red, Arp3 orange, ArpC1 green, ArpC2 light blue, ArpC3 violet, ArpC4 dark blue, and ArpC5 yellow. Actin is shown in gray. The numbering of the mother (M#) and daughter (D#) filament subunits is also indicated. Coloring was applied via the color zone command in ChimeraX in a 5.5 Å radius around each C-alpha atom of the underlying model, which was positioned in a 10 Å radius to a C-alpha atom of its putative interactor. **b** Surface representation of the interaction of ArpC2 and ArpC4 with the mother filament subunits M6 and M7 (the interaction of ArpC2 with M5 is omitted for clarity). **c** The protrusion helix of ArpC1 fitted into its density close to subdomains 1 and 3 of M4. **d** Surface representation of the interactions of Arp3 and ArpC3 with the mother filament. Note the cavity below Arp2, where no contacts between the Arp2 subunit and the mother filament are observed. ArpC3 acts as a linker between Arp2 and the mother filament. **e** Surface representation of the interaction between Arp2, Arp3, and the first two subunits of the daughter filament.

does not form any contacts with the mother filament. Instead, a density close to subdomains 1 and 3 of M4 fits to the previously proposed interaction of the ArpC1 protrusion helix (residues 297–305) with the mother filament[11,13] (Fig. 3a, c). The binding site of the ArpC1 protrusion helix on M4 closely overlaps with the reported hydrophobic interface on F-actin of Lifeact[39,40], a 17 amino acid peptide commonly employed for F-actin labeling[41] (Supplementary Fig. 7). This observation warrants further work to better understand if use of Lifeact changes Arp2/3 complex binding to the mother filament and hence activation.

Arp3 establishes contacts with three mother filament subunits (M4–M6), mostly via its D-loop and the base of subdomain 4 (Fig. 3a, d). Arp2 is not forming any contact with the mother filament. A cavity below Arp2 (Fig. 3d and Supplementary Movie 3) highlights the overall loose contacts between the Arp2/3 complex and M4 and M5. This is in line with our observation that formation and maintenance of a stable actin filament Arp2/3 complex branch junction do not require significant changes in actin subunits of the mother filament. This indicates that the mother filament does not need to adapt an unfavorable high-energy state to be primed for attachment of the Arp2/3 complex to its side[13], which might have implications for debranching or competition of actin regulatory factors with the Arp2/3 complex for the same binding site on the mother filament.

As described previously, ArpC3 is positioned to bind Arp3 and the mother filament, establishing small interfaces with actin subunits M2 and M4 (ref. [13]). In addition, in our model ArpC3 also acts as a bridge between Arp2 and the mother filament, where it provides further stabilization to the short-pitch conformation, since Arp2 has no direct contacts with the mother filament (Fig. 3a, d). This central role of ArpC3 in the formation and maintenance of the branch is in accordance with previous studies showing that ArpC3 KO mice and *S. pombe* are not viable[42,43]. ArpC5 does not form any contact with either the mother or the daughter filament, in line with previous reports suggesting its role to be mainly regulatory[30].

Arp2 and Arp3 are the only two subunits forming interfaces with the daughter filament (Fig. 3a, e), confirming that the stability of connecting the activated Arp2/3 complex to the newly growing daughter filament is solely accomplished by actin-like interfaces[13,35].

Overall, we note that most of the differences that can be observed between our branch junction model and the previous model by Rouiller et al.[13] most likely derive from the substantially lower resolution of the older structure, from which their fit was derived. This did not allow visualizing the conformational changes of the subcomplexes consisting of ArpC2, ArpC3, and Arp3 and ArpC4, ArpC1, ArpC5, and Arp2, respectively, and

hence limited the accuracy of fitting. Instead ArpC1, ArpC2, ArpC4, and ArpC5 where fit without modification, leading to an overall incorrect positioning of the complex on the mother filament (Supplementary Fig. 5c). Hence, while Rouiller et al. described the same mother actin filament subunits to be in contact with the Arp2/3 complex in the branch junction as we do, we observe a very distinct binding of the Arp2/3 complex subunits to these individual mother filament subunits. Correspondingly, the surface area covered by the Arp2/3 complex on the mother filament in the MD-refined model of the branch junction (~6000 Å$^2$)[16], which is based on the model published by Rouiller et al.[13] was significantly overestimated, when compared to a surface area of ~3100 Å$^2$ in our model.

**Arp2/3 complex activation and branch junction stabilization**. Arp2/3 complex activation during branch junction formation is associated with ATP binding[44] and requires the concerted action of two WCA binding events on two distinct sites on the Arp2/3 complex, specifically on Arp2–ArpC1 and Arp3 (refs. [6,8,45,46]). We asked if the described binding sites are accessible in our structure and if so, whether or not their location within the branch junction allows deriving further information about branch junction formation. A recent single-particle cryo-EM study of the inactive Arp2/3 complex bound to two WCA peptides showed that their C-helices interacted with structurally similar regions in both Arp2–ArpC1 and Arp3 (ref. [8]). Previously it was also suggested that for Arp2/3 complex activation, NPF binding to the Arp3 binding site requires the Arp3 C-terminal tail to be displaced from its position in the inactive conformation of the Arp2/3 complex, releasing its autoinhibitory function[8,44] and also eventually allowing the D-loop of the first daughter filament actin subunit to bind the Arp3-barbed end[35]. This agrees with our structure where no density corresponding to the C-terminal tail in its inactive conformation is present. Instead, our structure contains a density that places the Arp3 C-terminus in a position, where it is flipped toward ArpC4 and Arp2 (Fig. 4a).

In the inactive complex and the recent in vitro structure of the Dip1-activated *S. pombe* Arp2/3 complex, the ArpC5 N-terminus has been shown to bind the side of subdomain 3 of Arp2 (refs. [11,30,35]). In our structure, no density for this interaction can be observed (Fig. 4b). It is therefore possible that upon branch junction formation, the ArpC5 N-terminus disengages from this binding site on the Arp2/3 complex. In our structure, faint densities were appearing between two strongly positively charged regions of Arp2 (involving residues K299, K336, K339, K341, and R343) and ArpC1 (R94, R97, H136, K138, and K139) some of which have been implicated in binding the D/E loop in the WCA motif of NPFs[8]. As NPFs dissociate from the Arp2/3 complex upon branch junction formation[14,47], we assumed it to be unlikely that these densities can be attributed to residual NPF branch junction binding.

Instead, the length of the linker connecting the ArpC5 N-terminus to its central helical core and the size of the empty density in our branch junction structure would allow the N-terminus to be fitted in this empty density between ArpC1 and Arp2 (Fig. 4b). Specifically, the negatively charged region of the N-terminus of ArpC5 (D15, D17, E18, Y19, and D20) provides neutralization to the strongly positively charged region between ArpC1 and Arp2 (Fig. 4c, d). It is possible that via this binding, the ArpC5 N-terminus could aid in stabilizing the Arp2/3 complex in its active conformation after NPF release.

Recently, the mammalian Arp2/3 complex subunit isoforms ArpC1a, ArpC1b, ArpC5, ArpC5L were shown to assemble into functionally different complexes in defined combinations, resulting in significantly changed actin network dynamics, partially caused by their varying interaction with the NPF cortactin and the branch junction severing factor coronin 1b (ref. [48]). The binding site of the ArpC5 N-terminus partially overlaps with the previously proposed binding site of cortactin[7]. The differential regulatory activity of ArpC5 isoforms could be caused by their varying binding strength of their N-terminus to the active conformation of the Arp2/3 complex, playing a regulatory role in interactions with cortactin, specifically selecting for Arp2/3 complexes of ArpC1/ArpC5 subunit isoform composition[49].

In summary, our structure provides a more detailed understanding of the structural transformations and interactions being formed by the Arp2/3 complex upon branch junction formation. Our actin filament Arp2/3 complex branch junction model also poses exciting new questions and can serve as a platform for obtaining an even more thorough understanding of the mechanisms and function of this crucial actin filament nucleator, by guiding mutagenesis and biochemistry experiments. Obtaining structures from lamellipodia of migrating cells offers great potential for combining in situ structural biology experiments of regulatory factors of the actin cytoskeleton with their dynamic studies combining cell biology and genetics. Specifically, the contextual information determined by STA can allow studying the distribution, orientation, and structure of branch junctions within the lamellipodium in more detail. Combined with classification approaches, this could reveal if the structure varies depending on the localization in cells and correlates with the presence or absence of co-factors at the front or the back of a lamellipodium. Continued developments in cryo-EM hardware and software will allow obtaining even higher-resolution structures of the Arp2/3 complex within cellular environments, from both wild-type cells and cells lacking specific Arp2/3 complex subunit isoforms or regulatory proteins.

## Methods

**Cell culture**. Wild-type *Mus musculus* NIH-3T3 (RRID:CVCL_0594) fibroblast cells (kindly provided by Michael Sixt, IST Austria) were cultured in Dulbecco's modified Eagle's medium (DMEM GlutaMAX, ThermoFischer Scientific, #31966047), supplemented with 10% (v/v) fetal bovine serum (ThermoFischer Scientific, #10270106) and 1% (v/v) penicillin–streptomycin (ThermoFischer Scientific, #15070063). Cells were incubated at 37 °C and 5% $CO_2$.

Prior to Rac1Q61L transfection (plasmid kindly provided by Vic Small[24]) using Lipofectamine LTX with Plus Reagent (ThermoFischer Scientific, #15338030), NIH-3T3 cells were seeded at 75% confluency in a six-well plate and incubated at 37 °C and 5% $CO_2$ for 4 h. The primary transfection mix consisting of 2 μg of plasmid DNA encoding for Rac1Q61L, 200 μl of DMEM, and 2 μl of Plus Reagent was incubated at RT for 10 min. A total of 2 μl of LTX reagent was added and the mix was incubated at for another 30 min. The transfection mix was added dropwise to the cells. Cells were incubated at 37 °C and 5% $CO_2$ for 16 h prior to trypsinization, and seeding onto 200 mesh gold holey carbon grids (R2/2-2 C; Quantifoil Micro Tools), which were placed in 3D printed grid holders[50]. Prior to seeding, grids were glow discharged in an ELMO glow discharge unit (Cordouan Technologies) for 2 min and subsequently coated with 50 μg/ml fibronectin (Sigma-Aldrich, #11051407001) in PBS for 1 h at RT. Cells were allowed to settle on grids for 4 h at 37 °C and 5% $CO_2$ before the culture medium was exchanged with three washing steps with DMEM. Starvation was conducted at 37 °C and 5% $CO_2$ for 3 h before the cells were extracted and fixed according to a published protocol[24]. In detail, for extraction and fixation, grids were retrieved from grid holders and placed in a 50 μl drop of cytoskeleton buffer (10 mM MES, 150 mM NaCl, 5 mM EGTA, 5 mM glucose, and 5 mM $MgCl_2$, pH6.2) with 0.75% Triton X-100 (Sigma-Aldrich, #T8787), 0.25% glutaraldehyde (Electron Microscopy Services, #E16220), and 0.1 μg/ml phalloidin (Sigma-Aldrich, #P2141) and incubated for 1 min at RT. Grids were postfixed in a 50 μl drop of cytoskeleton buffer containing 2% glutaraldehyde and 1 μg/ml phalloidin for 15 min at RT, before being subjected to vitrification.

**Cryo-electron microscopy**. Samples were vitrified using a Leica GP2 plunger (Leica Microsystems) set to 4 °C and 80% humidity. After transfer into the blotting chamber, excess fixation solution was manually blotted off and 3 μl of a solution of 10 nm colloidal gold coated with BSA in PBS was added onto the grids. The grids were then vitrified in liquid ethane (−185 °C) after backside blotting (3 s), using the blotting sensor of the Leica GP2. Samples were stored under liquid nitrogen conditions until imaging.

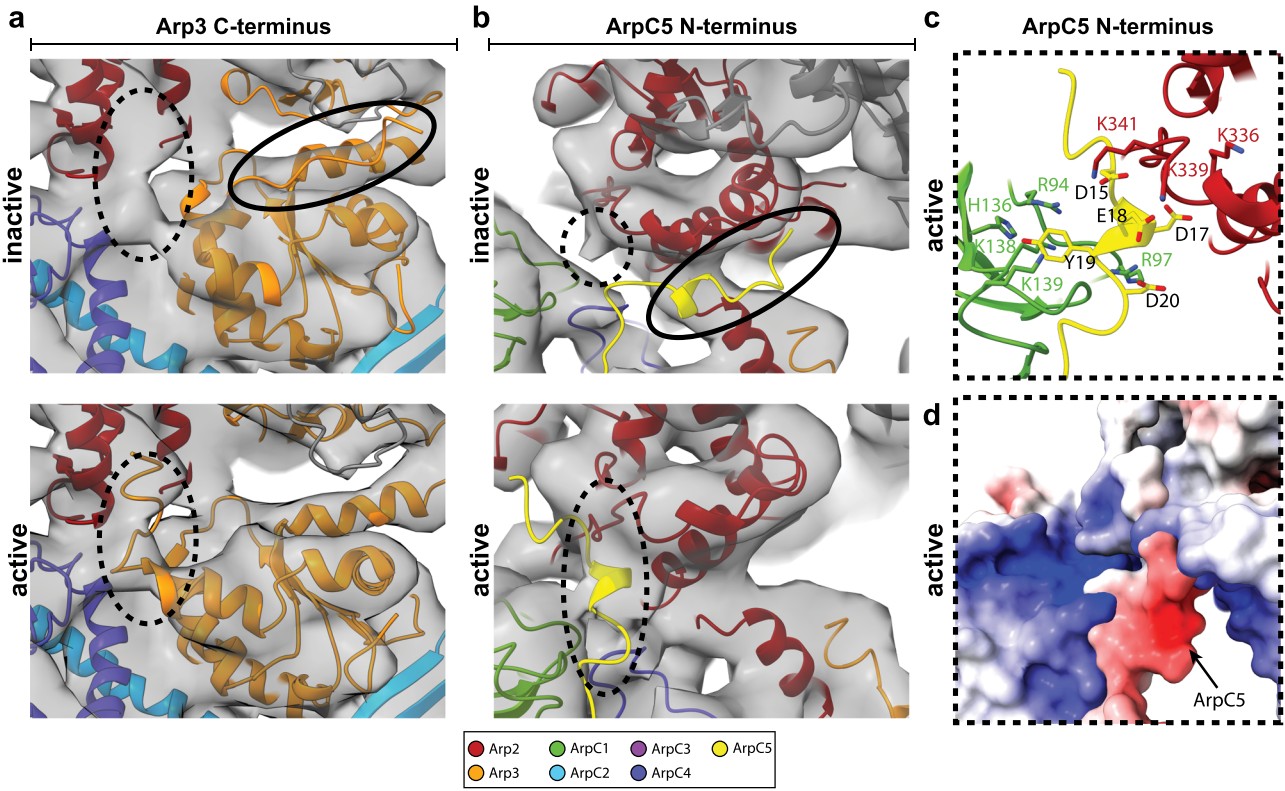

**Fig. 4 Structural changes in Arp3 and ArpC5 upon branch junction formation. a** Comparison of the conformation of the Arp3 C-terminal tail in the inactive and active conformation. No density is observed for the C-terminal tail of Arp3 in its inactive conformation (top). Instead, the C-terminal tail can flip toward Arp2 and ArpC4, where it is accommodated by an empty density present in the branch junction structure (bottom). **b** No density is observed for the ArpC5 N-terminus at its binding side in the inactive complex (top). Instead, the ArpC5 N-terminus can be fitted into a density between ArpC1 and Arp2 (bottom). **c** Positively charged residues in ArpC1 and Arp2 could coordinate the negatively charged N-terminus of ArpC5. **d** Electrostatic potential map of the area shown in **c**, with blue color indicating positive and red color indicating negative potentials. Subunit colors are annotated in the figure with Arp2 being red, Arp3 orange, ArpC1 green, ArpC2 light blue, ArpC3 violet, ArpC4 dark blue, and ArpC5 yellow. Actin is shown in gray.

Cryo-electron tomograms were acquired on a Thermo Scientific Titan Krios G3i TEM equipped with a BioQuantum post-column energy filter and a K3 camera (Gatan), using the SerialEM package[51]. Low- and medium-magnification montages were acquired for search purposes, and for defining areas of interest for subsequent high-resolution tomography data acquisition, respectively. Gain references were collected prior to data acquisition. Microscope and filter tuning were performed using SerialEM and DigitalMicrograph (Gatan) software, respectively. The slit width of the filter was set to 20 eV. Tilt series were acquired with a dose-symmetric tilt scheme[52] ranging from −60° to 60° with a 2° increment and a nominal defocus ranging from −1.75 to −5.5 μm. The nominal magnification was 42,000×, resulting in a pixel size of 2.137 Å. Individual tilt images were acquired as 11,520 × 8184 pixel super-resolution movies of seven frames. The calculated cumulative dose was 170 e/Å². Data were acquired over three acquisition sessions contributing 38, 12, and 81 tilt series, respectively, for a total of 131 tilt series, keeping above described acquisition parameters constant.

**Image processing**. Super-resolution movies were aligned on-the-fly during data acquisition, using the SerialEMCCD frame alignment plugin. Tilt series were automatically saved as 2×-binned (2.137 Å/px) mrc stacks. These stacks were used for image processing during template matching and principal component analysis (PCA)-based classification. CTFFIND4 (ref. [53]) was used to perform CTF estimation on each tilt individually. Images were low-pass filtered according to their cumulative electron dose. The appropriate filters were calculated using an exposure-dependent amplitude attenuation function and published critical exposure constants[54]. Prior to further processing, poor quality tilt images caused for example by grid bars blocking the beam at high tilt angles were removed. For preprocessing of tilt series (tilt stack sorting, removal of bad tilts, and exposure filtering), the tomoman software package was used (available via doi:10.5281/ZENODO.4110737). Tilt-series alignment of the exposure-filtered tilt images was done using the IMOD software package[55].

Initial processing steps including template matching, PCA-based classification, and STA were performed, using the Dynamo package[56] up to the generation of the reference for STA in RELION[57] (Supplementary Fig. 2).

To generate a starting reference for template matching 1549 branch junctions were manually picked from 37 8×-binned tomograms (17.096 Å/px), using the 3Dmod functionality of the IMOD software package. No angles were assigned to the manual picked positions. Cubic subvolumes with an approximate side length of 700 Å were extracted from 2×-binned tomograms and aligned against a structure of the branch junction derived previously from negative stain tomograms[24]. Alignment was done over five rounds, using the internal binning in Dynamo to resample subvolumes to 8.548 Å/px.

The resulting average was then band-pass filtered (100–40 Å) and used for template matching branch junctions in the entire dataset consisting of 131 tomograms. For cross-correlation calculation during template matching a mask consisting of two cylinders (both with a 140 Å radius) covering the branch, mother filament and the daughter filament were applied. Full 360° angular scanning during template matching was performed around all three axes with a sampling step of 10°.

False-positive cross-correlation peaks (i.e., from areas containing gray value outliers) were removed. Subsequently, the 300 positions with the highest cross-correlation value per tomogram were considered for further processing. For PCA classification in Dynamo, the corresponding 39,300 subvolumes were extracted from 8×-binned tomograms, split into three groups of equal size and processed in parallel to allow for faster computation. Pairwise cross-correlation calculations were performed for all particles within each group before PCA was conducted. Ten eigenvolumes were calculated and a subset of them was chosen to be employed for separating the particles into ten classes. Class averages containing only actin filaments were discarded, and class averages exhibiting equally strong densities for the whole branch junction, the mother and daughter filament were included for further processing. To this end, the remaining particles from the three classification groups were merged again, resulting in a total number of 17,302 subvolumes. The subvolumes of the complete dataset were subjected to one round of bin 8 alignment (Supplementary Fig. 2), and averaging in Dynamo to provide a reference for classification and STA in RELION[58].

The following processing steps were performed in Warp 1.0.7 (ref. [59]), M 1.0.9 (ref. [22]), and RELION 3.08 (ref. [57,60]).

In Warp super-resolution frames were binned (resulting pixel size was 2.137 Å) for frame alignment and defocus estimation of individual tilts. For tilt-series alignment, the same tilts and alignment parameters as determined in IMOD were employed. Defocus parameters were then again determined for whole tilt series. Coordinates of the particles determined from the Dynamo PCA calculation were employed for the extraction of 17,146 subvolumes into cubic subvolumes of 240 voxels at a pixel spacing of 2.137 Å and the corresponding 3D CTF/wedge models, which also consider radiation damage by accumulated electron dose.

One round of RELION 3D classification was performed, resulting in 14,296 subvolumes. This dataset was then subjected to RELION 3D auto-refine using the average determined in Dynamo (low-pass filtered to 40 Å), resulting in a resolution of 11.9 Å at the 0.143 criterion. Particles were automatically distributed into even and odd subsets during the RELION workflow. Subsequently, multiparticle refinement was performed in M. Tilt series were refined using image warp with a 9 × 6 grid and volume warp with a 4 × 6 × 2 × 10 grid. Particle poses and angles were refined for one temporal sampling point. These settings were kept for all following refinements in M. Subvolumes were re-extracted from the refined tilt series and aligned with RELION 3D auto-refine, using the result of the previous iteration filtered to 40 Å as reference. This cycling between Warp, RELION, and M was performed for a total of three rounds ending on the final iteration in M. RELION post-processing was applied to the resulting half-maps yielding a final structure at 9.0 Å resolution at the 0.143 FSC criterion (Supplementary Fig. 2). For B-factor sharpening, an empirically determined B-factor of −50 allowed to optimally visualize structural details in the EM density, without causing artificial discontinuous densities or sharp edges within the structure. Masks employed for resolution estimation encompassed the Arp2/3 complex and all actin subunits contacting it. Since the actin filaments extended to the box edge, we first masked in Dynamo around the region of interest (i.e., the branch junction and surrounding actin subunits). Then RELION was used to generate a mask around this region of interest, low-pass filtering the mask to 15 Å, extending it by five voxels and applying a soft edge of ten voxels (Supplementary Fig. 2).

**Model fitting and data analysis**. The crystal structure of the Arp2/3 complex with ATP bound to Arp2 and Arp3 (pdb 1TYQ)[12], and the model derived from the single-particle cryo-EM structure of aged, nucleotide-bound, and phalloidin-stabilized F-actin (pdb 6T20)[28] were used to generate a model of the actin filament Arp2/3 complex branch junction.

All subunits of the Arp2/3 complex and individual actin subunits from pdb 6T20 (11 in total, 8 within the densities of the mother filament and 3 within the densities of the daughter filament) were placed individually using the rigid-body fitting option in UCSF Chimera[61]. Due to the increased flexibility, as suggested by the lower resolution in our map, for the N-terminal region of ArpC5, rigid-body fitting only considered residues 69–151 of this subunit.

In most crystal structures of the Arp2/3 complex, subdomains 1 and 2 of Arp2 are not resolved, except for a structure of GMF-bound Arp2/3 complex[9]. In order to generate a complete model of Arp2 for fitting into our EM density map, we generated a composite Arp2 model consisting of subdomains 1 and 2 from pdb 4JD2 (GMF-bound Arp2/3 complex), and subdomains 3 and 4 from pdb 1TYQ (not having a GMF interaction in the original crystal structure). The exact composition of the resulting model, which was then rigid-body fitted is reported in Supplementary Table 2.

For ArpC1, the "protrusion" helix formed by the residues 297–305 and the surrounding linker region is not present in the 1TYQ model. The helix itself and residues 306–309 were imported from pdb 1K8K[11], and placed in an empty density at the surface of the actin subunit M4. Merging of the two models and adding residues connecting them, was performed in Coot[62]. The source of the primary structure elements of the resulting model is given in (Supplementary Table 2). Smaller gaps in the models of other individual subunits were bridged by adding the missing residues in Coot and N- or C-terminal regions were trimmed if no fitting density could be found. Supplementary Table 2 refers to the implemented changes. Phalloidin models were removed and the 4-methyl-histidines in the actin structure (pdb 6T20) at position 73 were replaced by histidines. The complete assembly containing all Arp2/3 subunits, actin subunits, and remaining ligands was merged into a single pdb file. Coot was then used to move residues within clashing areas and to release entanglements between chains. The resulting model was associated to the sharpened map in ChimeraX[63] and hydrogens were added using the addh command. The ISOLDE plugin[29] was employed to restrict the positions of ligands (ATP and Mg$^{2+}$/Ca$^{2+}$ ions from the original models), ArpC5 and the actin subunits M1, M8, and D3. It was further used to manually move parts of the Arp2 domain 2 (residues 37–56), Arp3 C-terminus (residues 405 onward), and to perform a MD simulation for the whole model to allow for an initial flexible fitting into the corresponding densities. In a next step, ChimeraX was employed to exchange truncated residues present in the original models with their full-length counter parts (Supplementary Table 2). Using ISOLDE, smaller areas were simulated locally to manually alleviate clashes, improve fitting of sub chains, and correct for highly improbable geometries. For this, secondary structure restraints were applied whenever necessary to keep α-helices from deteriorating, and positional restraints were applied to keep the actin structures from occupying densities associated with phalloidin. After regional changes were implemented, the whole model was activated again via ISOLDE and cooled down to 0 K. After all

ligands had been removed, the quality of the model was validated using MOLPROBITY[64], as it is integrated in the Phenix Comprehensive validation (Cryo-EM) tool[65] (Supplementary Table 3).

Visualization and model analysis were performed with UCSF ChimeraX. For the comparisons of inactive and active conformation shown in Fig. 2, maps were generated via the molmap command in ChimeraX at described resolutions from pdb 1TYQ and our branch junction model, respectively (1TYQ was modified to contain all residues present in our model).

In order to accurately calculate the branch junction angle, vectors representing the mother filament and the daughter filament were derived from fitting a line between the C-alpha atoms of residues Met269 of M1 and M8 (vector M1to8) for the mother filament, and residues Met269 for D1 and D3 (vectorD1to3) of the daughter filament, respectively. Met269 was chosen due to its proximity to the central axis of the actin filament. The branch angle was then calculated via (1).

$$\Theta = \arccos \frac{\textbf{vectorM1to8} \cdot \textbf{vectorD1to3}}{|\textbf{vectorM1to8}||\textbf{vectorD1to3}|} \quad (1)$$

RMSD calculations were performed considering only the C-alpha atoms of the models. Models were aligned and compared in Chimera using the matchmaker and the RMSD command, respectively. PDB2PQR[66] and APBS[67] were employed to calculate electrostatic potential maps, which were used to color surfaces in ChimeraX.

The surface area calculation between the Arp2/3 complex and the mother filament in our model and the model derived by MD simulation[16] (both with hydrogens included) was performed employing the "measure buriedarea" command in ChimeraX, using a 1.5 Å probe and otherwise standard settings.

**Reporting summary**. Further information on research design is available in the Nature Research Reporting Summary linked to this article.

## Data availability
Data supporting the findings of this manuscript are available from the corresponding author upon reasonable request. A reporting summary for this article is available as a Supplementary Information file. The EM density map of the actin filament Arp2/3 complex branch junction and one representative tomogram have been deposited in the EMDB (https://www.ebi.ac.uk/pdbe/emdb/) under accession numbers EMD-11869 and EMD-11870, respectively. The model of the actin filament Arp2/3 complex branch junction has been deposited in the PDB (https://www.rcsb.org/) under accession code PDB 7AQK. The models we used for fitting into our structure (PDB 6T20 for actin, PDB 1TYQ, PDB 1K8K, and PDB 4JD2 for the Arp2/3 complex) and the model of the Dip1-activated Arp2/3 complex PDB 6W17 used for comparison are available from the PDB (https://www.rcsb.org/).

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

## Acknowledgements

This research was supported by the Scientific Service Units (SSUs) of IST Austria through resources provided by Scientific Computing (SciComp), the Life Science Facility (LSF), the BioImaging Facility (BIF), and the Electron Microscopy Facility (EMF). We

also thank Dimitry Tegunov (MPI for Biophysical Chemistry) for helpful discussions about the M software, and Michael Sixt (IST Austria) and Klemens Rottner (Technical University Braunschweig, HZI Braunschweig) for critical reading of the manuscript. We also thank Gregory Voth (University of Chicago) for providing us the MD-derived branch junction model for comparison. The authors acknowledge support from IST Austria and from the Austrian Science Fund (FWF): M02495 to G.D. and Austrian Science Fund (FWF): P33367 to F.K.M.S.

## Author contributions

Conceptualization, supervision, project administration, and funding acquisition: F.K.M.S.; methodology, validation, formal analysis, data curation, writing—original draft, and visualization: F.F. and F.K.M.S.; software: F.F. and W.W.; investigation: F.F., G.D., and V.-V.H.; and writing—review and editing: F.F., G.D., V.-V.H., W.W. and F.K.M.S.

## Competing interests

The authors declare no competing interests.
