## [Peer Review File · Nature Communications]

REVIEWER COMMENTS

Reviewer #1 (Remarks to the Author):

In this manuscript, Fäßler et al. determine the in situ structure of actin filament branch junctions to subnanometer resolution.

They are able to unambiguously fit atomic models of all components of the junctions. Interaction partners are defined based on proximity, and the arrangement of the Arp2/3 dimer is assessed. Based on their results, the authors propose a model for branch formation and stabilisation. This model is consistent with previous biochemical data but differs significantly to that obtained using in vitro reconstitutions of actin branches.

The work is technically excellent, and it provides significant novel insight into actin branching mechanisms. I have a list of minor comments which should be addressed prior to publication, listed below.

1. The word 'novel' should be removed from the title
2. In the first paragraph of the intro references are missing (lines 31,35 and 37)
3. Lines 42-43: it is unclear what 'WCA', 'W', and 'CA' refer to.
4. Line 61: 'accurate structure' should be 'accurate model'.
5. Lines 90-92: it is unclear how Fig S1 and Video 1 show that extraction and fixation don't change the ultrastructure. Different evidence would be needed to support this statement (e.g. images before and after), or the sentence needs to be rephrased.
6. It is not obvious to the general reader what 'extraction' is, please clarify.
7. Line 100: models don't have a high or low resolution, please rephrase.
8. Line 115: it is unclear how the angle was measured. One would think that depending on how lines are drawn the measurement would differ quite significantly. The authors need to specify in the methods how they've measured the angle and consider the possibility that the variations reported in the literature might be due to different ways of measuring.
9. Figure 3a. The legend reports that the point of view is that of the daughter filament, but I still find it confusing to understand. It looks to me as though the interfaces on the left 2 panels are shown after 'opening' the complex (like matching points on opposite pages of an open book). If so, it would be good to have a schematic representation of this. If this is the case, I wonder whether one of the two panels on the right side is upside-down?
10. Line 169: the word 'color' seems out of place.
11. Line 196: the authors need to explain what the 'barbed' end is. Reference to barbed and pointed ends is also made in the video but it won't be clear to people outside of the actin field.
12. Methods: the authors mention half maps for the postprocessing. I assume dataset splitting was done automatically as part of the relion auto-refinement routine while the dynamo processing for the starting reference was done with the whole dataset? It would be helpful to state explicitly.
13. Methods: the authors mention using the 'tomoman' software. This requires a reference or an explanation.
14. Methods: the authors should discuss how they determined the sharpening level to use.
15. Deposition: it looks from the validation report that no masks or half maps were deposited. I recommend these are added to the deposition.
16. I also have a general comment/curiosity: I wonder whether any useful information could be extracted from the positions and orientations of the aligned subtomograms. I guess that in combination with a filament tracking software to identify continuous actin filaments it would be possible to assess the distance and relative orientation/direction between subsequent branching points. One might find this isn't random, and gain some insight into regulation? I do not suggest the authors do this for this paper, but it might be something to consider for future studies.

Sincerely
Giulia Zanetti

Reviewer #2 (Remarks to the Author):

Please find enclosed the revision of the manuscript NCOMMS-20-36561 entitled "Novel cryo-electron tomography structure of Arp2/3 complex in cells reveals mechanisms of branch formation"
The authors used cryo-electron tomography to generate a model of branched actin filament mediated by the Arp2/3 complex.
They proposed that the model obtained from their cellular analysis is different from the model made based on in vitro reconstitution.

Overall, I think this is an interesting study. However, I have few comments that the authors should address before publication.

1) The title is misleading. The data presented here provide a structure of the branched actin filament mediated by the Arp2/3 complex in a cellular context. The mechanism of branch formation (i.e.: side binding interaction to the mother filament, activation by WASP family of proteins, and binding to actin monomers) is somehow a speculative model based on the data, so I will not emphasize the mechanism in the title.

2) In the abstract the authors (last sentence) mentioned that their model differs from in vitro branch junction model. Maybe, the variability is coming from the fact that in a cellular context actin filament can bind to other actin binding proteins or are subject to forces during lamellipodium growth. This is of course not the case in vitro. The authors should at least discuss why they believe the difference comes from the mechanism of branch formation and not alternative effects as mentioned above.

3) The data are based on subtomogram averaging, it will be interesting to discuss if the structure obtained varies depending of the localization in the cells (front of the lamellipodium, further back) where the tomograms were taken.

Reviewer #3 (Remarks to the Author):

Fäßler et al. Novel cryo-electron tomography structure of Arp2/3 complex in cells reveals mechanisms of branch formation.

This paper presents the highest resolution structure available for the branch junction between actin filaments formed by Arp2/3 complex. Remarkably, this has been done by cryo-electron tomography of whole cells and subtomogram averaging, which limits the number of particles available for reconstruction, but still is a substantial, welcome improvement on previous work from 2008 using negatively stained specimens.

Overall, this is an excellent paper. Below I suggest a change in the balance between the sections; more on the structural details and comparisons with previous work and less on speculation about the activation mechanism, which is really not informed by the new structure.

Detailed comments

Line 30: "Spontaneous actin filament polymerization is an energetically unfavourable process" is a true statement but mainly it is kinetically unfavorable.

Line 43: Ref 2 is very outdated. Much new work has been done by the Dominguez, Nolen, Pollard and

Rosen labs among others since 2009.

Fig 1: The reconstruction is very impressive. The rendering of this figure and the other figures is outstanding, making it easy to appreciate the new information. Video 3 is particularly helpful for illustrating how the subunits fit into the map density. The resolution of secondary structure is outstanding, but the complete absence of side chains causes me to caution the authors about over-interpreting the structural details.

Line 103: The following statement “without requiring any large-scale modifications of their tertiary structure” in the following is ambiguous, “allowed an unambiguous placement of available high-resolution models of the individual Arp2/3 complex subunits and actin monomers into our density, without requiring any large-scale modifications of their tertiary structure.” Actin monomers and polymerized actin subunits have different conformations, so be clear that you used a subunit from a filament with bound phalloidin and partially dissociated phosphate. Using an ADP-actin filament subunit would have been more appropriate. Explain in the main text whether the individual subunits of inactive Arp2/3 complex built into the density or the whole inactive complex?

Lines 104, 293 and elsewhere: the actin molecules in filaments are usually called subunits, not monomers. If you do not want to call them subunits, molecules would be better than monomers, since they certainly do not have the same conformation as monomers.

Line 104: I wonder why the structure of Arp2/3 complex (PDB: 1TYQ) was used for fitting into the EM reconstruction. This structure has bound ATP, while Arp2/3 complex in the branch junction is likely to have bound ADP. Furthermore, the resolution of 1TYQ is only 2.55 Å, while the original, nucleotide-free structure had a resolution of 2.0 Å. Explain in the main text how you dealt with the lack of density for subdomains 1 and 2 of Arp2 in the 1TYQ structure.

Line 135: the rotation that repositions Arp2 next to Arp3 in this branch junction structure and the Shaaban (2020) structure with Dip1 looks similar to the rotation discovered by steered molecular dynamics simulations by Dalhaimer (Biophys. J. 99:2568-2576. PMID: 20959098). A detailed comparison of the new structure with that earlier work should be included in this paper. The authors of ref 9 did not know about this rotation, so they fit the crystal structure into the low resolution EM density as follows: “To generate this conformation of the Arps, to move the Arp2/3 complex into the density, and to avoid steric clashes with the rest of the complex, we made the following changes: (1) we moved Arp2 next to Arp3 in a short-pitch helix dimer; (2) we closed the nucleotide binding clefts of both Arps; and (3) we twisted subdomains 3 and 4 of both Arps by ~15 Å relative to subdomains 1 and 2.” I always thought that step 1 (picking up and moving Arp2) was arbitrary, and now we see that it seems to have led to inaccurate positioning of Arp2 relative to the mother filament. The two new lines of evidence (this paper and Shaaban) for the rotation mechanism provide a simple explanation for the differences between this new model and that in ref 9. I would explain this to the readers.

Line 138: The statement “other activating mechanisms must exist” in the following is ambiguous: “this structural change is induced by binding of Dip1 to ArpC4, other activating mechanisms must exist to induce this conformation in the branch junction, as the ArpC4 binding site is occupied by the mother filament (see below).” The mother filament is well documented to be required to activate most types of Arp2/3 complex, so the mother filament is the obvious candidate for the “other activating mechanism” but is not mentioned.

Lines 147-187: The new information about contacts between Arp2/3 complex and the mother actin filament is the most important part of the paper. I had trouble understanding how the new model compares with that in ref 9. The reader needs an introductory overview before presenting a detailed comparison. The overview might note if both models have Arp2/3 complex docked similarly on the mother filament such that it contacts the same subunits. I tried to figure out these contacts by comparing Fig. 3A in this paper with Fig 4B in ref 9, but could not be sure about any details. Are the

mother filament actin subunits named the same way in the two papers?

After this overview, I would describe in detail the contacts between Arp2/3 complex and the mother filament in the two models. If space limitations are a problem, I would reduce the size of the section on "Arp2/3 complex activation and branch junction stabilization," which is more speculative, since the new structure has no direct information about the activation pathway.

Line 168: provide a reference for "proposed interaction of the ArpC1 protrusion helix (residues 297-305) with the mother filament."

Line 173: why is "color" included in the phrase "actin monomers M2 color and M4?"

Line 182: I would include a reference in the following "not need to adapt an unfavourable high-energy state to be primed for attachment of the Arp2/3 complex to its side (9)."

Line 192: "two distinct sites on the Arp2/3 complex, specifically on Arp2-ArpC1 and Arp3 [33]." Ref 33 was a pioneering study, but has been superseded by new work from several labs.

Lines 222-227: This is the authors' hypothesis for activation. The only reference is to 36, which deals with what happens after activation and is inappropriate. Instead, the authors should consider the spectroscopic and crosslinking data in papers from the Nolen, Dominguez and Pollard labs that address the activation mechanism. Those papers concluded that NPF binding drives the conformational change part way toward the active complex. Budding yeast Arp2/3 complex is exceptional in that NFP binding suffices for slow activation, but Arp2/3 complex from other organisms depends on binding to the side of the mother filament for activation. This important feature of the activation mechanism is ignored here.

Line 227: Start a new paragraph with "Recently, the mammalian Arp2/3 complex subunit..." This is a different topic from the activation mechanism and should have its own paragraph.

Line 237: The material starting with "activation of the Arp2/3 complex is associated with" does belong with the material on activation.

Supplemental materials

The numbers of significant figures in some parts of the supplemental materials is excessive.

Line 140: "cryo-em structure of an ATP and Phalloidin bound actin filament (pdb 6T20)." I think this is wrong. The ATP was hydrolyzed during polymerization and much of the phosphate dissociated.

Tom Pollard

We thank the reviewers for their valuable and encouraging comments and the editor for allowing us to revise our manuscript. As requested, we have addressed the reviewer's comments and questions and have laid out our responses in detail below.

Specifically, we have changed the focus of the results section to more strongly emphasize the structural comparison with the previous branch junction model. Correspondingly, we have removed the part of the manuscript where we had speculated about the activation mechanism. We have also added Supplementary Fig. 7 and a short section to the results (Page 5, Lines 199-203) where we describe the observation that the ArpC1 protrusion helix shares the same binding site as Lifeact, a commonly used F-actin label.

Please also find all changes in a separately uploaded version of the manuscript with tracked changes.

REVIEWER COMMENTS

Reviewer #1 (Remarks to the Author):

In this manuscript, Fäßler et al. determine the in situ structure of actin filament branch junctions to subnanometer resolution.

They are able to unambiguously fit atomic models of all components of the junctions. Interaction partners are defined based on proximity, and the arrangement of the Arp2/3 dimer is assessed. Based on their results, the authors propose a model for branch formation and stabilisation. This model is consistent with previous biochemical data but differs significantly to that obtained using in vitro reconstitutions of actin branches. The work is technically excellent, and it provides significant novel insight into actin branching mechanisms. I have a list of minor comments which should be addressed prior to publication, listed below.

1. The word 'novel' should be removed from the title

We have changed the title as suggested by reviewer 1 and 2.

The new title is "Cryo-electron tomography structure of Arp2/3 complex in cells reveals new insights into branch formation"

2. In the first paragraph of the intro references are missing (lines 31,35 and 37)

As requested, we have added references to the first paragraph of the introduction

3. Lines 42-43: it is unclear what 'WCA', 'W', and 'CA' refer to.

We have adapted the corresponding sentence to clarify that 'WCA' is the combination of WASP homology 2 (W), connector (C), and acidic (A).

4. Line 61: 'accurate structure' should be 'accurate model'.

This has been corrected.

5. Lines 90-92: it is unclear how Fig S1 and Video 1 show that extraction and fixation don't change the ultrastructure. Different evidence would be needed to support this statement (e.g. images before and after), or the sentence needs to be rephrased.

We have rephrased the statement on extraction and fixation.

Page 3, Line 88-90

"In agreement with previous observations, branch junctions were clearly retained within lamellipodia after extraction and fixation (Supplementary Fig. 1, Supplementary movie 1)."

6. It is not obvious to the general reader what 'extraction' is, please clarify.

We have rephrased the sentence, so that the effect of the extraction is clearly stated:

Page 3, Line 84-88

"In order to further optimize conditions for image processing we used established extraction and fixation protocols involving the actin-stabilizing toxin phalloidin to remove the plasma membrane while stabilizing actin filaments, allowing for a better visualization of actin filaments and bound complexes (24, 26)."

7. Line 100: models don't have a high or low resolution, please rephrase.

We have removed the term 'high-resolution' in this context.

8. Line 115: it is unclear how the angle was measured. One would think that depending on how lines are drawn the measurement would differ quite significantly. The authors need to specify in the methods how they've measured the angle and consider the possibility that the variations reported in the literature might be due to different ways of measuring.

We have added a description on how the angle was measured to the *Methods* section.

Methods, Page 11/Line 476-483:

"In order to accurately calculate the branch junction angle, vectors representing the mother filament and the daughter filament were derived from fitting a line between the C_{alpha} atoms of residues Met269 of M1 and M8 (vector M1to8) for the mother filament and residues Met269 for D1 and D3 (vector D1to3) of the daughter filament, respectively. Met269 was chosen due to its proximity to the central axis of the actin filament. The branch angle was then calculated via

$$\theta = \arccos \frac{\text{vectorM1to8} \cdot \text{vectorD1to3}}{|\text{vectorM1to8}| |\text{vectorD1to3}|}$$

"

We have also modified the sentence describing the branch angle in the main text.

Page 4, Line 121-125:

"We determined the angle of the branch junction in our structure to be 71 degrees, which is in agreement with some, but different to other previously reported in vitro and in situ studies, which showed branch junction angles ranging from 67 to ~80 degrees (13, 15, 16, 24, 30–32), a variation that could be partially due to potentially different approaches for angle measurement".

9. Figure 3a. The legend reports that the point of view is that of the daughter filament, but I still find it confusing to understand. It looks to me as though the interfaces on the left 2 panels are shown after 'opening' the complex (like matching points on opposite pages of an

open book). If so, it would be good to have a schematic representation of this. If this is the case, I wonder whether one of the two panels on the right side is upside-down?

We have updated Figure 3a and its respective figure legend to make the viewing angles clearer. Specifically, we have changed the annotations in Figure 3a, and rotated the right most depiction of the Arp2/3 complex model by 180 degrees to ease orientation within the figure.

10. Line 169: the word 'color' seems out of place.

We have removed the word color

11. Line 196: the authors need to explain what the 'barbed' end is. Reference to barbed and pointed ends is also made in the video but it won't be clear to people outside of the actin field.

Barbed end and pointed end are established terms in the field of actin and are regularly used in the context of describing sites on Arp2 or Arp3. We have therefore kept this specific reference to the barbed end. We have, however, changed now the annotations in the movies and the respective movie legends to refer to the sides of the Arp2/3 complex facing the mother or daughter filament.

12. Methods: the authors mention half maps for the postprocessing. I assume dataset splitting was done automatically as part of the RELION auto-refinement routine while the dynamo processing for the starting reference was done with the whole dataset? It would be helpful to state explicitly.

No dataset splitting was performed during the bin8 alignment in Dynamo, as this alignment step was only used for generating the starting reference for RELION. The reviewer is also correct in that the dataset splitting was performed automatically within the RELION auto-refinement routine.

We have added the information about whether the particles were split into half-sets into the respective Methods section:

Methods, Page 9/ Line 392-394

"The subvolumes **of the complete dataset** were subjected to one round of bin 8 alignment (Supplementary Fig. 2) and averaging in Dynamo to provide a reference for classification and subtomogram averaging in RELION (58)."

Methods, Page10, Line 407-408:

"Particles were automatically distributed into even and odd subsets during the RELION workflow."

13. Methods: the authors mention using the 'tomoman' software. This requires a reference or an explanation.

The tomoman software is available via the Zenodo repository <https://zenodo.org/record/4110737#.X486Q-17ljF> and has an associated DOI:

10.5281/zenodo.4110737. We have updated the manuscript to include the respective citation.

14. Methods: the authors should discuss how they determined the sharpening level to use. We have added a sentence explaining the rationale behind our empirical B-factor determination:

Methods, Page 10/Line 416-418

“For B-factor sharpening, an empirically determined B-factor of -50 allowed to optimally visualize structural details in the electron microscopy density without causing artificial discontinuous densities or sharp edges within the structure. “

15. Deposition: it looks from the validation report that no masks or half maps were deposited. I recommend these are added to the deposition.

The final half maps and the mask used for FSC determination have been added to the deposition.

The deposited structure, the model and a representative tomogram can be accessed via accession codes EMD-11869, PDB 7AQK and EMD-11870 respectively.

16. I also have a general comment/curiosity: I wonder whether any useful information could be extracted from the positions and orientations of the aligned subtomograms. I guess that in combination with a filament tracking software to identify continuous actin filaments it would be possible to assess the distance and relative orientation/direction between subsequent branching points. One might find this isn't random, and gain some insight into regulation? I do not suggest the authors do this for this paper, but it might be something to consider for future studies.

The reviewer makes a very valuable point here. In another ongoing project we are indeed actively studying the distribution of branch junctions within lamellipodia of cells under varying conditions.

We have added a sentence to the discussion that mentions the possibility of exploring branch junction distribution (and associated structural changes) within lamellipodia:

Page 7/Line 293-297

“Specifically, using the contextual information determined by subtomogram averaging can allow studying the distribution, orientation and structure of branch junctions within the lamellipodium in more detail. Combined with classification approaches, this could reveal if the structure varies depending on the localization in cells and correlates with the presence or absence of co-factors at the front or the back of a lamellipodium.”

Please also see our response to reviewer #2, who made a similar comment.

Reviewer #2 (Remarks to the Author):

Please find enclosed the revision of the manuscript NCOMMS-20-36561 entitled "Novel cryo-electron tomography structure of Arp2/3 complex in cells reveals mechanisms of branch formation"

The authors used cryo-electron tomography to generate a model of branched actin filament mediated by the Arp2/3 complex.

They proposed that the model obtained from their cellular analysis is different from the model made based on *in vitro* reconstitution.

Overall, I think this is an interesting study. However, I have few comments that the authors should address before publication.

1) The title is misleading. The data presented here provide a structure of the branched actin filament mediated by the Arp2/3 complex in a cellular context. The mechanism of branch formation (i.e.: side binding interaction to the mother filament, activation by WASP family of proteins, and binding to actin monomers) is somehow a speculative model based on the data, so I will not emphasize the mechanism in the title.

As suggested, we have changed the title to not refer to the mechanism of branch formation. Please also see our response to reviewer 1.

2) In the abstract the authors (last sentence) mentioned that their model differs from *in vitro* branch junction model. Maybe, the variability is coming from the fact that in a cellular context actin filament can bind to other actin binding proteins or are subject to forces during lamellipodium growth. This is of course not the case *in vitro*. The authors should at least discuss why they believe the difference comes from the mechanism of branch formation and not alternative effects as mentioned above.

We thank the reviewer for this comment. We have realized that our abstract was ambiguous in its statement, where we make the comparison of our model to the *in vitro* model by Rouiller *et al.*

We indeed believe that there might be several reasons why our cellular branch junction model differs from the previous model, which was obtained via negative stain electron tomography of *in vitro* generated branch junctions.

The most likely reason for the differences is the lower resolution of the *in vitro* branch junction model. This has resulted in less precise fitting of the Arp2/3 complex subunits into the EM-density.

In the course of revising our manuscript, we have also changed the emphasis of our manuscript to the structural description of our model. Hence, following also the advice of reviewer #3 we have now extended manuscript with a more detailed comparison of our model and the model published by Rouiller *et al.*

In order to avoid the impression that we assume that the main differences between our model and the previously published model is predominantly caused by the *in vitro* system, we have removed the term *in vitro* from the abstract.

3) The data are based on subtomogram averaging, it will be interesting to discuss if the structure obtained varies depending of the localization in the cells (front of the lamellipodium, further back) where the tomograms were taken.

We thank the reviewer for this suggestion. We are already actively following this up in another project, where we are establishing workflows to classify branch junctions according to their lamellipodial location and to identify any potential structural changes. Please also see our response to reviewer #1, who had a similar comment.

Reviewer #3 (Remarks to the Author):

Fäßler et al. Novel cryo-electron tomography structure of Arp2/3 complex in cells reveals mechanisms of branch formation.

This paper presents the highest resolution structure available for the branch junction between actin filaments formed by Arp2/3 complex. Remarkably, this has been done by cryo-electron tomography of whole cells and subtomogram averaging, which limits the number of particles available for reconstruction, but still is a substantial, welcome improvement on previous work from 2008 using negatively stained specimens.

Overall, this is an excellent paper. Below I suggest a change in the balance between the sections; more on the structural details and comparisons with previous work and less on speculation about the activation mechanism, which is really not informed by the new structure.

We thank the reviewer for these encouraging comments. As suggested, we have now aimed to change the emphasis of our manuscript towards the structural details and their comparison with previous work.

Detailed comments

Line 30: “Spontaneous actin filament polymerization is an energetically unfavourable process” is a true statement but mainly it is kinetically unfavorable.

We have changed the sentence to use the word ‘kinetically’ instead of ‘energetically’.

Line 43: Ref 2 is very outdated. Much new work has been done by the Dominguez, Nolen, Pollard and Rosen labs among others since 2009.

We have updated the manuscript to include more appropriate references.

Fig 1: The reconstruction is very impressive. The rendering of this figure and the other figures is outstanding, making it easy to appreciate the new information. Video 3 is

particularly helpful for illustrating how the subunits fit into the map density. The resolution of secondary structure is outstanding, but the complete absence of side chains causes me to caution the authors about over-interpreting the structural details.

We thank the reviewer for this comment and also agree that the limited resolution warrants caution in interpretation. We have tried avoiding any interpretations on the side chain level. To this end we also deposit our model of the actin filament Arp2/3 complex branch junction as backbone trace model only.

We have also added an additional statement to the main text.

Page 3/Line 116-118:

“Overall, our model reveals the conformation of in total 18 protein subunits (the heptameric Arp2/3 complex and 11 actin filament subunits), however the resolution of our structure did not allow deriving unambiguous side chain information.”

Line 103: The following statement “without requiring any large-scale modifications of their tertiary structure” in the following is ambiguous, “allowed an unambiguous placement of available high-resolution models of the individual Arp2/3 complex subunits and actin monomers into our density, without requiring any large-scale modifications of their tertiary structure.” Actin monomers and polymerized actin subunits have different conformations, so be clear that you used a subunit from a filament with bound phalloidin and partially dissociated phosphate. Using an ADP-actin filament subunit would have been more appropriate. Explain in the main text whether the individual subunits of inactive Arp2/3 complex built into the density or the whole inactive complex?

In order to choose the appropriate actin model for fitting into our electron microscopy density, we have determined the initial rigid body fits of actin subunits from various available models. We found that fitting an ADP-actin subunit of pdb 6DJO resulted in a marginally lower cross-correlation score compared to the actin subunit from a filament with bound phalloidin and partially dissociated phosphate (pdb 6T20). As our reconstruction contains phalloidin we decided to use a subunit from pdb 6T20.

As recommended, we have now rephrased this paragraph to specifically state the origin of the models and that the chains were all fit individually.

Page 3, Line 100-103:

“For this, the subunits of the inactive Arp2/3 complex (pdb 1TYQ) and 11 actin molecules derived from a nucleotide and phalloidin bound, aged actin filament (pdb 6T20) were all individually rigid-body fitted into the electron microscopy density.”

Lines 104, 293 and elsewhere: the actin molecules in filaments are usually called subunits, not monomers. If you do not want to call them subunits, molecules would be better than monomers, since they certainly do not have the same conformation as monomers.

We have replaced the term ‘monomer’, when referring to actin subunits within filaments, throughout the text.

Line 104: I wonder why the structure of Arp2/3 complex (PDB: 1TYQ) was used for fitting into the EM reconstruction. This structure has bound ATP, while Arp2/3 complex in the branch junction is likely to have bound ADP. Furthermore, the resolution of 1TYQ is only 2.55 Å, while the original, nucleotide-free structure had a resolution of 2.0 Å. Explain in the main text how you dealt with the lack of density for subdomains 1 and 2 of Arp2 in the 1TYQ structure.

The resolution of our map is too low to visualize the nucleotide binding state. To this end, we decided to refine the model of Arp3 that showed the highest initial rigid body fit score. Hence, the choice for pdb 1TYQ over the original (nucleotide-free) Arp2/3 complex model (pdb 1K8K) or the Arp2/3 complex bound to ADP (pdb 1U2V) was based on assessing the initial rigid body fits of the Arp3 subunits of these different models into the density of Arp3 in our EM reconstruction. Here, pdb 1TYQ showed the highest correlation score (we note, however, that the difference between pdb 1TYQ and pdb 1U2V was small). Due to the limited resolution of our map we have also decided to not include nucleotides in the final (deposited) model.

The information on how we have added subdomains 1 and 2 to the incomplete Arp2 of pdb 1TYQ has been included in the updated paragraph.

Page 3/Line103-105

“As subdomains 1 and 2 of Arp2 were not present in pdb 1TYQ, the corresponding subdomains from a GMF-bound Arp2/3 complex (9) (pdb 4JD2) were used to produce a chimeric Arp2 model.”

Line 135: the rotation that repositions Arp2 next to Arp3 in this branch junction structure and the Shaaban (2020) structure with Dip1 looks similar to the rotation discovered by steered molecular dynamics simulations by Dalhaimer (Biophys. J. 99:2568-2576. PMID: 20959098). A detailed comparison of the new structure with that earlier work should be included in this paper. The authors of ref 9 did not know about this rotation, so they fit the crystal structure into the low resolution EM density as follows: “To generate this conformation of the Arps, to move the Arp2/3 complex into the density, and to avoid steric clashes with the rest of the complex, we made the following changes: (1) we moved Arp2 next to Arp3 in a short-pitch helix dimer; (2) we closed the nucleotide binding clefts of both Arps; and (3) we twisted subdomains 3 and 4 of both Arps by ~15 Å relative to subdomains 1 and 2.” I always thought that step 1 (picking up and moving Arp2) was arbitrary, and now we see that it seems to have led to inaccurate positioning of Arp2 relative to the mother filament. The two new lines of evidence (this paper and Shaaban) for the rotation mechanism provide a simple explanation for the differences between this new model and that in ref 9. I would explain this to the readers.

We have included a comparison with the work of Dalhaimer, Biophys J. 2010 into the revised manuscript, when describing the repositioning of Arp2.

Page 4/Line 140-151:

“In both structures of the activated Arp2/3 complex, as well as in a model derived by steered MD simulations (35), ArpC2 and ArpC4 define the centre of rotation and translation of two subcomplexes consisting of ArpC2, Arp3, ArpC3 and ArpC4, ArpC5, ArpC1, Arp2 that move against each other to form the short pitch conformation (Supplementary Movies 4 and 5). In the case of unbranched filaments, nucleated by the Dip1-activated Arp2/3 complex, this structural change is induced by binding of Dip1 to ArpC4. In the branch junction this conformation is most likely induced by the actin mother filament, which forms an extensive interface with the ArpC2 and ArpC4 subunits (see below). Indeed, the mother filament has been shown to be required for the activation of most types of the Arp2/3 complex (13, 33, 37) and its role in promoting the conformational changes in the Arp2/3 complex to reposition Arp2 has also been suggested in a study using steered MD simulations (35).”

We have also extended the comparison of our structure with the work by Rouiller et al, as suggested (see our response to one of the comments below).

Line 138: The statement “other activating mechanisms must exist” in the following is ambiguous: “this structural change is induced by binding of Dip1 to ArpC4, other activating mechanisms must exist to induce this conformation in the branch junction, as the ArpC4 binding site is occupied by the mother filament (see below).” The mother filament is well documented to be required to activate most types of Arp2/3 complex, so the mother filament is the obvious candidate for the “other activating mechanism” but is not mentioned.

We have updated the revised manuscript to now describe that the mother filament is required for Arp2/3 complex activation in most cases. Please see our response to the comment above.

Lines 147-187: The new information about contacts between Arp2/3 complex and the mother actin filament is the most important part of the paper. I had trouble understanding how the new model compares with that in ref 9. The reader needs an introductory overview before presenting a detailed comparison. The overview might note if both models have Arp2/3 complex docked similarly on the mother filament such that it contacts the same subunits. I tried to figure out these contacts by comparing Fig. 3A in this paper with Fig 4B in ref 9, but could not be sure about any details. Are the mother filament actin subunits named the same way in the two papers? After this overview, I would describe in detail the contacts between Arp2/3 complex and the mother filament in the two models. If space limitations are a problem, I would reduce the size of the section on “Arp2/3 complex activation and branch junction stabilization,” which is more speculative, since the new structure has no direct information about the activation pathway.

As suggested by the reviewer we have added an introductory paragraph on the Arp2/3 complex mother filament interaction as published by Rouiller et al, followed by a more exhaustive comparison of the contacts between the Arp2/3 complex and the mother filaments in both models.

The numbering direction of the mother filament subunits in our manuscript follows the convention described by *Pfaendtner and colleagues* (reference 16 in our manuscript), where the numbering of the mother filament subunits starts from the pointed end. In contrast, *Rouiller et al*, count the mother filament subunits from the barbed end. We deemed the convention described by *Pfaendtner et al*, to be more appropriate as the daughter filament subunits are also counted from the pointed end.

Since our reconstruction of the branch junction only shows good density for 8 mother filament subunits, our count deviates from the MD-derived model by *Pfaendtner et al*.

Despite these differences, subunit M4 in our model corresponds to subunit M4 in Rouiller et al.

We understand that the differences in naming the mother filament subunits between the models are unfortunate, but we hope that the more detailed comparison between our model and the previously published model by *Rouiller et al* makes it easier for the reader to follow.

Line 168: provide a reference for “proposed interaction of the ArpC1 protrusion helix (residues 297-305) with the mother filament.”

We have added the appropriate reference to this statement.

Line 173: why is “color” included in the phrase “actin monomers M2 color and M4?”

The word ‘color’ in this position has been removed

Line 182: I would include a reference in the following “not need to adapt an unfavourable high-energy state to be primed for attachment of the Arp2/3 complex to its side (9).”

We have added the appropriate reference to this statement.

Line 192: “two distinct sites on the Arp2/3 complex, specifically on Arp2-ArpC1 and Arp3 [33].” Ref 33 was a pioneering study, but has been superseded by new work from several labs.

As suggested, we have added additional citations at this place in the manuscript.

Lines 222-227: This is the authors’ hypothesis for activation. The only reference is to 36, which deals with what happens after activation and is inappropriate. Instead, the authors should consider the spectroscopic and crosslinking data in papers from the Nolen, Dominguez and Pollard labs that address the activation mechanism. Those papers concluded that NPF binding drives the conformational change part way toward the active complex. Budding yeast Arp2/3 complex is exceptional in that NFP binding suffices for slow activation, but Arp2/3 complex from other organisms depends on binding to the side of the mother filament for activation. This important feature of the activation mechanism is ignored here.

In the course of revising the manuscript we have removed this section from the text.

Line 227: Start a new paragraph with “Recently, the mammalian Arp2/3 complex subunit...”

This is a different topic from the activation mechanism and should have its own paragraph. We have introduced a line break at the indicated site.

Line 237: The material starting with “activation of the Arp2/3 complex is associated with” does belong with the material on activation.

This part of the manuscript has substantially changed upon revision. The information on the activation of the Arp2/3 complex has now been moved to the correct position in the text.

Supplemental materials

The numbers of significant figures in some parts of the supplemental materials is excessive. We have reduced the number of digits in Table S3 where appropriate.

Line 140: “cryo-em structure of an ATP and Phalloidin bound actin filament (pdb 6T20).” I think this is wrong. The ATP was hydrolyzed during polymerization and much of the phosphate dissociated.

This sentence has been corrected, as suggested.

Methods; Page 10/Lines 426-429:

*“The crystal structure of the Arp2/3 complex with ATP bound to Arp2 and Arp3 (pdb 1TYQ) (12) and the model derived from the single particle cryo-EM structure of **aged, nucleotide-bound and phalloidin-stabilized F-actin** (pdb 6T20) (38) were used to generate a model of the actin filament Arp2/3 complex branch junction. ”*

Additional comments for the revised manuscript:

We have noticed an inaccuracy in our previous description of the methods, and hence we have also updated the *Methods* section on Image Processing. Specifically, we have removed the part where we describe the CTF-correction via NovaCTF. NovaCTF was only used in early attempts and trials of processing our data, in which we tried to identify the best image processing approaches.

In the final workflow, which resulted in the 9 Å structure of the branch junction, NovaCTF was not included. Correspondingly, we have also updated Supplementary Fig. 2.

REVIEWERS' COMMENTS

Reviewer #1 (Remarks to the Author):

I am satisfied with the authors' revisions, and I am happy to recommend publication.

Reviewer #3 (Remarks to the Author):

The authors responded to my review very thoroughly and thoughtfully. The revised ms is greatly improved, so I can recommend it for publication.

Tom Pollard